# Rapid measurement of inhibitor binding kinetics by isothermal titration calorimetry

Justin M. Di Trani [1], Stephane De Cesco[1], Rebecca O'Leary[1], Jessica Plescia[1], Claudia Jorge do Nascimento[1,2], Nicolas Moitessier[1] & Anthony K. Mittermaier[1]

Although drug development typically focuses on binding thermodynamics, recent studies suggest that kinetic properties can strongly impact a drug candidate's efficacy. Robust techniques for measuring inhibitor association and dissociation rates are therefore essential. To address this need, we have developed a pair of complementary isothermal titration calorimetry (ITC) techniques for measuring the kinetics of enzyme inhibition. The advantages of ITC over standard techniques include speed, generality, and versatility; ITC also measures the rate of catalysis directly, making it ideal for quantifying rapid, inhibitor-dependent changes in enzyme activity. We used our methods to study the reversible covalent and non-covalent inhibitors of prolyl oligopeptidase (POP). We extracted kinetics spanning three orders of magnitude, including those too rapid for standard methods, and measured sub-nM binding affinities below the typical ITC limit. These results shed light on the inhibition of POP and demonstrate the general utility of ITC-based enzyme inhibition kinetic measurements.

[1] Department of Chemistry, McGill University, Montreal, QC H3A 0B8, Canada. [2] Institute of Biosciences, Federal University of the State of Rio de Janeiro, Urca, 22290-240 Rio de Janeiro, Brazil. Correspondence and requests for materials should be addressed to A.K.M. (email: anthony.mittermaier@mcgill.ca)

There is mounting evidence that the efficacy of a therapeutic is closely related to the kinetics of interactions with its target[1], particularly its residence time. Systemic drug concentrations fluctuate according to administration and excretion/metabolism and substrates of inhibited enzymes tend to accumulate. Long-residence times allow targets to remain inhibited even when the systemic drug concentrations drop[2–6] or substrate concentrations rise to a level that would otherwise overwhelm the effect of the drug[7]. On the other hand, molecules with slow association kinetics are disfavored in typical drug screens with short pre-incubation steps[8], and potentially efficacious molecules may be missed altogether unless care is taken. This has prompted an interest in structure–kinetics relationships (SKR) to better understand the relationship between the structures of small molecule drug candidates and their kinetic properties[9–11].

Enzyme kinetic studies typically employ spectroscopic[12,13], chromatographic[3,13], or electrophoretic[13] techniques to monitor the concentrations of products or substrates as a function of time, thereby yielding rates of catalysis. To measure the strength of inhibition, $K_i$ or $IC_{50}$, the enzyme (E) is allowed to equilibrate thoroughly with an inhibitor (I), such that concentration of the inhibited complex (EI) can be considered time invariant. To characterize the inhibitor association ($k_{on}$) and dissociation ($k_{off}$) rate constants, the pre-equilibration time with the inhibitor is varied[14], or substrate and product concentrations are measured while the concentration in EI gradually changes due to inhibitor binding and release[15]. Using traditional enzyme assays to probe inhibition kinetics has several drawbacks. For instance, experiments must be repeated multiple times with the different pre-equilibration delays and/or inhibitor concentrations. Also, it can be difficult to detect small changes in catalytic rate by simply measuring substrate and concentrations over time. New biophysical methods, to quickly and efficiently assess the binding kinetics of drug candidates, are needed to improve screening and optimization efforts and to better understand the fundamental mechanisms underlying enzyme inhibition.

Enzyme kinetics can also be characterized by isothermal titration calorimetry (ITC), which measures the heat generated by catalysis following the rapid mixing of enzyme and substrate[16]. An ITC experiment consists of making a series of automated injections from a syringe into a sample cell and monitoring the subsequent heat flow. There are many advantages to ITC-based enzyme measurements: they can be performed under dilute, physiological solution conditions, even those that are spectroscopically opaque[17]. The approach is completely general since most of the chemical reactions produce or consume heat; ITC can be applied equally well to virtually any enzyme[16], and does not require the development of a customized assay based on fluorogenic or colorigenic substrates, or the post-reaction separation of products and substrates by chromatography or electrophoresis[16,18]. Unlike standard spectroscopic measurements where enzyme, substrate, and inhibitor solutions are combined with delays of tens of seconds or more prior to the start of the measurement, ITC measures heat flow while the reagents are mixed rapidly with little dead time. Furthermore, in contrast to other techniques that infer rates of catalysis indirectly from the concentrations of substrates and products, ITC detects heat flow in real time, giving a direct readout of enzyme activity and how it varies in response to inhibitors. Despite the great potential of ITC to characterize the kinetics of enzyme inhibition to our knowledge no study has employed it in this manner till date.

Here we present a pair of rapid, complementary ITC methods that simultaneously measure inhibitor association and dissociation rates and the inhibitory constant $K_i$, for enzyme inhibitors in an hour or less. We used these methods to characterize several

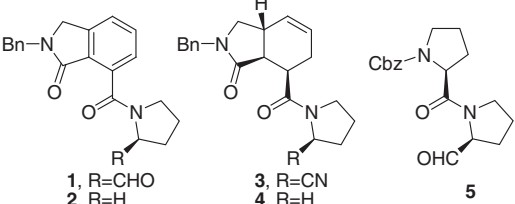

**Fig. 1** POP inhibitors employed in this study. Bn and Cbz indicate benzoyl and carboxybenzyl groups, respectively

covalent and non-covalent inhibitors (Fig. 1) of prolyl oligopeptidase (POP), a post-proline cleaving enzyme implicated in cancer and neurodegenerative disorders[19,20]. Compounds **2** and **4** bind non-covalently to POP, while **1**, **3**, and **5** form reversible covalent bonds with the catalytic serine in the POP active site via aldehyde (**1** and **5**) or nitrile (**3**) moieties. Covalent inhibitors are promising as long-acting drugs, while fine tuning the reactivity of the warhead offers an opportunity for optimizing kinetics. Relatively little is currently known about SKR for covalent inhibitors since they have historically been disfavored in drug development due to concerns regarding specificity and off-target effects. Nevertheless, many over-the-counter and blockbuster drugs are covalent[21,22], and the advantages of this class of compounds, including high specificity, potency, and favorable kinetics of inhibition, are increasingly being recognized[21,23]. The typically slow kinetics of covalent drugs makes them ideal candidates for developing new biophysical methods and elucidating new structure–activity relationships for the kinetics of inhibition. Application of the ITC kinetics methods presented here to POP inhibitors has provided proof-of-principle for this approach and has yielded insights into the fundamental mechanisms underlying covalent and non-covalent enzyme inhibition.

## Results

**Covalent inhibition**. One complication of covalent inhibitors is that their binding mechanisms are usually considered to have at least two distinct steps[12,13,24,25]. The enzyme and inhibitor first interact non-covalently (EI) and subsequently form a covalent bond (E−I, Eq. (1)).

$$E + I \rightleftharpoons EI \rightleftharpoons E - I. \tag{1}$$

Depending on the rates of covalent bond formation and association and dissociation of EI, the kinetics of inhibition can be either biphasic, characterized by rapid formation of EI followed by gradual conversion to E−I[12,24] or monophasic, such that they are indistinguishable from the kinetics of simple, one-step non-covalent inhibitors, as further discussed in Supplementary Methods[12,13]. Importantly, the time resolution of our ITC experiments allows us to discriminate between these two possibilities (monophasic versus biphasic inhibition).

**Kinetic ITC techniques**. ITC kinetic data for the POP enzyme associating with compound **1** and **4** (Fig. 2a–d) and dissociating from compound **1** and **4** (Fig. 2e–h) are shown in Fig. 2. In what follows, we will refer to these two types of experiments as kinetics of inhibition and kinetics of initiation. In both experiments, the heat flow or power (y-axis) is plotted as a function of time (x-axis). Exothermic and endothermic reactions deflect the ITC signal downward and upward, respectively. The power is linearly related to the enzyme velocity, with larger deflections corresponding to higher velocities. In the kinetics of inhibition experiments, the ITC cell contained POP and the substrate

thyrotropin-releasing hormone (TRH). POP cleaves TRH producing the free acid form of TRH and ammonia, as well as heat, which is detected by the ITC instrument (Supplementary Note 2). The rate of catalysis was initially constant giving a horizontal line. Compounds **1** and **4** were added to the cell in a series of four (Fig. 2a) and seven (Fig. 2c) injections. In each case the enzyme was increasingly inhibited and the power values shifted upward, since the rate of (exothermic) catalysis and downward deflection was reduced after each injection. As highlighted in Fig. 2b, d, this shift occurred gradually over tens to hundreds of seconds, which corresponds to the time required for compounds **1** and **4** to bind in the active site. Furthermore, each subsequent injection led to a smaller upward shift of the ITC signal, as the enzyme became increasingly saturated with inhibitor. The decrease in the sizes of the steps is related to the inhibition constant, $K_i$. Data for a series of injections were fitted simultaneously to yield $k_{on}$ and $K_i$ (Supplementary Methods). The fitted curves are in excellent agreement with the experimental ITC signal (Fig. 2b, d). The disassociation rate was then calculated as $k_{off} = k_{on} \times K_i$. In the kinetics of initiation experiments, the cell contained the substrate TRH and the syringe contained a solution of POP saturated with the compounds **1** (Fig. 2e, f) or **4** (Fig. 2g, h), which was added to

the cell in five or six injections. Immediately following each injection there was no change in the rate of catalysis in the sample cell as the added enzyme was fully inhibited. However, the large dilution (>20-fold) experienced by the injectant led to a net dissociation of the inhibitor and a gradual downward shift of the ITC signal as the freshly released enzyme began to act on the substrate (Fig. 2e, g). Each subsequent injection led to a smaller downward shift of the ITC signal, as the inhibitor accumulated in the sample cell and the net dissociation of each injection diminished. The decrease in the sizes of the steps is governed by the value of $K_i$. Data for the series of injections were fitted simultaneously to yield $k_{off}$ and $K_i$ (Supplementary Methods), giving excellent agreement (Fig. 2f, h). The association was then calculated as $k_{on} = k_{off}/K_i$. Note that the concentrations of enzyme are so low in these experiments (≈10 nM) that ITC detects only heats of catalysis, while heats of inhibitor/enzyme binding can be safely ignored (Supplementary Methods).

**Kinetic characterization of POP inhibitors by ITC.** In order to test the applicability of these ITC kinetic methods, they were applied to characterize the panel of POP inhibitors illustrated in Fig. 1. The extracted thermodynamic and kinetic values are listed

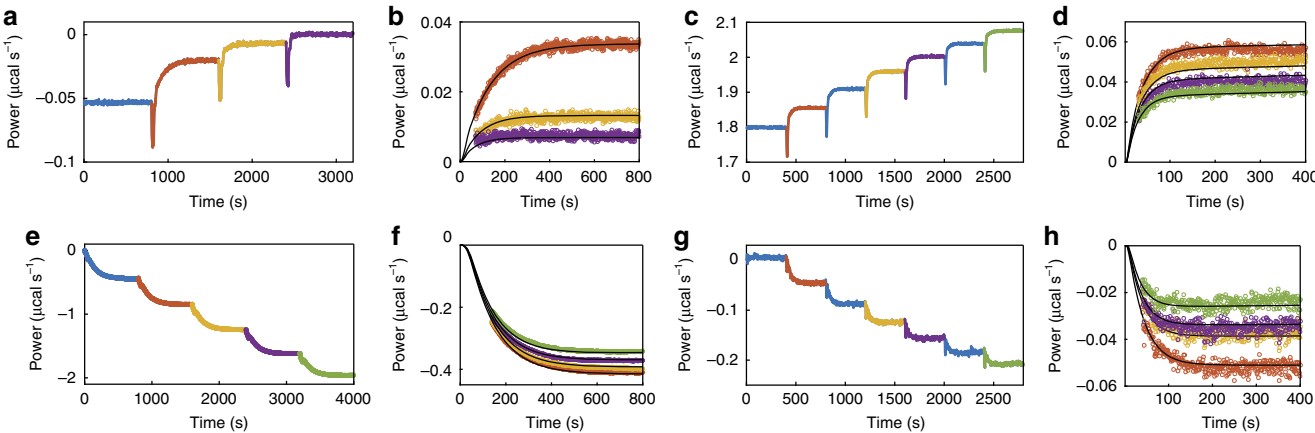

**Fig. 2** Kinetics of inhibition and initiation experiments. **a** Kinetics of inhibition experiment with compound **1** using a Malvern ITC-200 calorimeter. Compound **1** is titrated into cell containing POP and TRH. **b** Overlay of injections 2, 3, and 4 (orange, yellow, and purple circles) with fit (black line). **c** Kinetics of inhibition experiment with compound **4** using a Malvern ITC-200 calorimeter. Compound **4** is titrated into cell containing POP and TRH. **d** Overlay of injections 2, 4, 5, and 7 (orange, yellow, purple, and green circles) with fit (black line). **e** Kinetics of initiation experiment of compound **1** with a Malvern VP-ITC calorimeter. Compound **1** and POP are titrated in the cell containing TRH. **f** Overlay of injections 2, 3, 4, and 5 (orange, yellow, purpl,e and green circles) with fit (black lines). **g** Kinetics of initiation experiment with compound **4** using a Malvern ITC-200 calorimeter. Compound **4** and POP are titrated into cell containing TRH. **h** Overlay of injections 2, 4, 5, and 7 (orange, yellow, purple, and green circles) from **c** with fit (black lines)

**Table 1 Kinetic and thermodynamic parameters**

| Cpd | Experiment | $k_{on} \times 10^5\ M^{-1}\,s^{-1}$ | $k_{off} \times 10^{-4}\,s^{-1}$ | $K_i$ nM | [a]$\Delta H_{cat}$ kcal mol$^{-1}$ |
|---|---|---|---|---|---|
| **1** | ITC inhib. | 1.12 ± 0.02 | 43.1[b] ± 0.9 | 38.4 ± 0.4 | −6.69 ± 0.02 |
|  | ITC init. | 1.21[b] ± 0.01 | 82.1 ± 0.4 | 58.7 ± 0.5 | −5.86 ± 0.01 |
|  | UV-Vis | 1.43 ± 0.02 | 31.4 ± 0.3 | 947[b] ± 2 | — |
| **2** | ITC inhib. | — | — | 597 ± 28 | −6.79 ± 0.03 |
|  | ITC init. | — | — | 744 ± 16 | −10.88 ± 0.06 |
| **3** | ITC inhib. | 155.5 ± 0.4 | 9[b] ± 1.5 | 0.063 ± 0.01 | −9.91 ± 0.03 |
| **4** | ITC inhib. | 44 ± 1 | 249[b] ± 7 | 5.6 ± 0.1 | −5.33 ± 0.03 |
|  | ITC init. | 20[b] ± 1 | 261 ± 9 | 13.1 ± 0.4 | −7.19 ± 0.04 |
| **5** | ITC inhib. | 4.18 ± 0.04 | <1.05 | <2.5 | −7.97 ± 0.01 |
|  | NMR | 5.4 ± 0.7 | — | — | — |
|  | Lit[34] | 0.7 | 3 | 0.5 | — |

[a] $\Delta H_{cat}$ are floated in order account for uncertainties in enzyme concentration as the magnitude of the heat flow is directly proportional to $[E_0]\Delta H_{cat}$
[b] Calculated using $K_i = k_{off}/k_{on}$

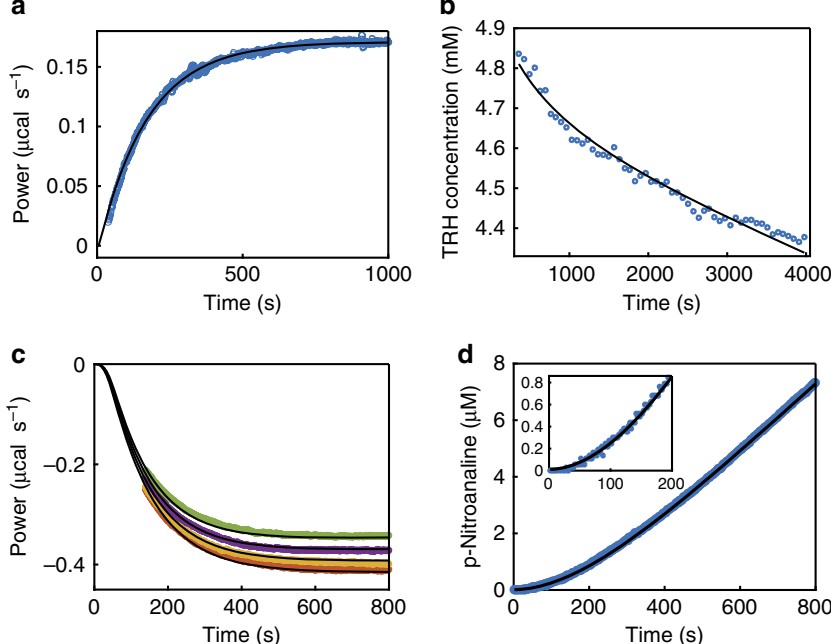

**Fig. 3** Validation using absorbance spectroscopy and ¹H NMR. **a** Inhibition experiment with compound **5** using a Malvern ITC-200 calorimeter. Compound **5** is injected into the cell containing POP and TRH (blue circles). **b** Inhibition with compound **5** using NMR. Compound **5** is added to POP and TRH. Intensity of peak for TRH was used to calculate a concentration (blue circles). **c** Initiation experiment with compound **1** using a Malvern ITC-200 calorimeter. POP and compound **1** are injected into the cell containing TRH (second injection; orange circles, third injection; yellow circles, fourth injection; purple circles, and fifth injection; green circles). **d** Initiation experiment with compound **1** using UV-Vis spectroscopy. POP and compound **1** are injected into buffer containing ZGP-pNA. The intensity of the peak at 405 nm was converted to a concentration of pNA (blue circles). Fits to experimental data are overlaid (black lines) using models presented in Supplementary Methods

in Table 1, while the raw fits are shown in Supplementary Figs. 7–17. Most experiments were performed in under an hour, and replicate experiments were highly reproducible. For compounds **1** and **4**, both the initiation and inhibition kinetics methods yielded the full set of $k_{on}$, $k_{off}$, and $K_i$, parameters, as described above. Notably the values obtained from the two methods are in the excellent agreement, differing by at most about a factor of two. Global fits of combined initiation and inhibition data sets are in good agreement with experiment signals and yield similar kinetic and thermodynamic parameters to the individual fits (Supplementary Note 1). For compound **2**, the affinity of the interaction was quite weak and the residence time quite short. Consequently, both association and dissociation processes went nearly to completion during each injection and kinetic parameters could not be extracted with great confidence. Nevertheless, both ITC methods yielded $K_i$ values which agreed well with each other. For compounds **3** and **5**, dissociation occurred very slowly compared to the rate of ITC baseline drift and $k_{off}$ could not be accurately measured using the initiation experiment (Supplementary Note 2). In both cases, however, the inhibition experiment provided well-defined $k_{on}$ values, illustrating the complementary nature of the two ITC approaches. In the case of **3**, this experiment also gave $K_i$. For **5**, the $k_{on}$ rate was very low (30-fold less than **3**), meaning that in order to achieve measurable kinetics a high concentration of inhibitor was used. This saturated the enzyme in a single step and only upper bounds for the $K_i$ and $k_{off}$ could be determined (Supplementary Note 2). It should be mentioned that for some covalent inhibitors of other enzymes, association kinetics is markedly biphasic, due to accumulation of non-covalent binding intermediates[12,24]. In our case, all association and dissociation traces are clearly monophasic, implying that non-covalent binding intermediates are not highly populated and that association and dissociation are well-described by the $k_{on}$ and $k_{off}$ rate constants (Supplementary Methods).

**Validation of ITC-based techniques**. The fact that the inhibition and initiation ITC kinetics experiments gave matching values of $k_{on}$, $k_{off}$, and $K_i$ gives us confidence in these methods, as the two types of measurements were performed completely independently. In order to further test their reliability, we selected representative inhibition and initiation experiments for validation using standard spectroscopic techniques. For compound **5**, we used NMR to measure the loss-of-enzyme activity after addition of the inhibitor. We mixed enzyme, substrate, and inhibitor, placed the sample in the spectrometer, and monitored a peak present in the ¹H NMR spectrum of the TRH substrate but not in that of the free acid product using the signal intensity to calculate the concentration of TRH as a function of time. The curvature of the resulting plot in Fig. 3b is proportional to the association rate. Note that the final non-zero slope of the curve in Fig. 3b is due to remaining uninhibited POP. Increasing the amount of **5** added would make this portion of the curve more horizontal, but would also increase the rate of inhibition (early curvature) beyond values accessible by this method, highlighting some of the challenges of measuring inhibition kinetics by non-ITC methods. The extracted value of $k_{on}$ (Table 1) is within error of that determined using the ITC inhibition experiment (Fig. 3a). For compound **1**, we used absorbance spectroscopy and a commercially available colorigenic substrate Cbz-Gly-Pro-pNA (ZGP-pNA[26]) to measure the rate at which the inhibited enzyme regained activity following dilution. POP was pre-equilibrated with a saturating concentration of inhibitor, diluted into a solution of ZGP-pNA, and the spectroscopic absorbance was used to calculate product concentration as a function of time (Fig. 3d). The slight curvature of the resulting plot (see inset) is proportional to the inhibitor dissociation rate, $k_{off}$. The fitted value (Table 1) is consistent (within a factor two) of the results of the ITC initiation experiment (Fig. 3c). It must be emphasized that, while we have used spectroscopic methods to validate the ITC kinetics measurements, the range of applicability

of the ITC kinetics experiments is far greater due to the shorter delay between mixing and detection and the greater sensitivity to changes in catalytic rate.

**SKR for POP inhibitors**. There is currently not much information available on how the structures of covalent inhibitors relate to their binding kinetic behavior, thus the data in Table 1 shed new light on this important relationship. Of particular interest is the effect of adding a reactive warhead on the association and dissociation rate constants of a given scaffold. **1** differs from the non-covalent inhibitor, **2** by virtue of an aldehyde moiety. This results in about a 10-fold increase in affinity, as expected, since the enzyme complex with **1** is stabilized by an additional covalent bond. The dissociation rate also shows a substantial decrease. For the non-covalent inhibitor **2**, $k_{off}$ is too rapid to measure, i.e., larger than about $0.1\,\mathrm{s}^{-1}$, while for the covalent inhibitor **1**, the dissociation rate is reduced to about $60 \times 10^{-4}\,\mathrm{s}^{-1}$, likely due to the added kinetic barrier of breaking a covalent bond. Similarly, **3** differs from **4** by a reactive nitrile group, and binds about 100-fold more tightly. This is largely due to the 25-fold decrease in the dissociation rate, similar to what was seen for **1** and **2**. Interestingly, the association rate for **3** is about 3.5-fold higher than that of the non-covalent **4**. This is unexpected, since the rate of non-covalent complex formation represents an upper bound for the kinetics of inhibition (Eq. S5). The nitrile moiety may somehow speed the $\mathrm{E} + \mathrm{I} \rightarrow \mathrm{EI}$ step although how this would occur is unclear. Alternatively, it is believed that the vast majority of molecular collisions between proteins and their ligands are unproductive, in the sense that the binding partners diffuse apart again before a tight complex can form[27]. The reaction of the nitrile group with the catalytic serine may effectively trap some collisions that would otherwise be unproductive, thereby enhancing the association rate. This possibility is provocative, as it would imply that the reaction pathway toward the covalently inhibited complex could circumvent the non-covalent intermediate, to some extent. Regardless of the underlying mechanism, these results suggest that the covalent warheads can have multiple beneficial effects on the kinetics of binding, simultaneously increasing the rate of association and the residence time.

## Discussion

There is a rich literature on the use of ITC to study enzyme kinetics[16,17], and to a lesser extent on the kinetics of ligand binding[28,29]. This study brings the power of ITC, as a near universal enzyme assay, to bear on the important problem of inhibitor dynamics. It must be emphasized that, although, we have focused on covalent inhibition here, these ITC techniques can be equally well applied to non-covalent inhibitors, for instance compounds **4** and **2**. We estimate that inhibitor residence times from roughly 10 s to 30 min, association rates from $10^3$ to $10^7\,\mathrm{M}^{-1}\,\mathrm{s}^{-1}$, and $K_i$ values down to about 1 pM are accessible using this approach under typical conditions (see Supplementary Note 2 for details and a practical guide to use these techniques). This work also demonstrates some specific advantages of ITC compared to existing methods. For instance, since ITC measures enzyme catalytic rates directly, the signatures corresponding to inhibition and initiation are much more pronounced than those of methods that rely on measuring substrate or product concentrations. This is well illustrated in Fig. 3, where ITC data show large exponential changes in heat flow (a, c), while the corresponding spectroscopic experiments (b, d) show only slight curvature. This provides an opportunity for deep mechanistic analysis, as deviations from first-order kinetics could be clear in ITC data and largely obscured in product or substrate concentration measurements. The information rich ITC data also allows full thermodynamic

and kinetic characterization in a single-hour-long experiment, simultaneously yielding $k_{on}$, $k_{off}$, and $K_i$. In contrast, spectroscopic techniques require multiple separate enzyme assays to be performed over a range of inhibitor concentrations to provide the same information. Furthermore, the ITC instrument rapidly mixes enzyme, substrate, and inhibitor solutions and records heat flow with a little dead time. It is therefore ideal for characterizing inhibitors that associate or dissociate rapidly. The most rapidly dissociating inhibitor in this study (compound **4**) has a residence time of just 40 s. We have recently used ITC to extract enzyme kinetics from reactions that go to completion in just 20 s[30], and anticipate that the inhibition kinetics on those timescales would be accessible as well. This is well beyond the limits of conventional benchtop spectroscopy and begins to approach the timescales of stop flow instruments. Finally, these methods also represent a novel approach for applying ITC to extremely tightly interacting inhibitors. In order to obtain reliable information, the concentration of enzyme in the ITC cell is ideally not more than 500-fold the equilibrium dissociation constant, which is generally taken as equal to the $K_i$[31]. Thus, very tight inhibitors require the use of very dilute enzyme solutions and produce extremely weak signals, if the heats of binding are measured directly, as is usually the case. This problem can be circumvented by using the higher concentrations of enzyme and performing competitive binding assays with a known ligand that binds weakly to the same site[32]. Our experiments provide an alternative by detecting the heat generated by catalysis. Even very low (1.32 nM) enzyme concentrations produce large heat signals due to multiple substrate turnovers. This allowed the measurement of a $K_i$ as low as 63 pM in this study (compound **3**), which would not have been attainable by standard ITC titration methods, with an estimated lower limit of about 1 pM. These ITC kinetics experiments thus provide a robust and versatile approach for measuring inhibitor binding kinetics and thermodynamics, with the potential to more clearly reveal how inhibitors, both covalent and non-covalent, dock against their targets, how they dissociate, and how this relates to their chemical structures.

## Methods

**POP purification**. Human prolyl oligo peptidase was purified as described in Supplementary Methods section[33] and dialyzed in a buffer containing 20 mM sodium phosphate, pH 8, 150 mM sodium chloride, and 10% (v/w) glycerol, prior to flash freezing in ~300 μL aliquots at ≈1–4 μM in liquid nitrogen and storing at −80 °C. Kinetic experiments were carried out with freshly thawed POP in the same buffer above with BSA (0.5 mg/mL) added to help stabilize POP. The substrate TRH was purchased from BACHEM international (product H-4915) and dissolved into the same buffer as POP for kinetics experiments.

**Experimental conditions**. All experiments were carried out in 20 mM sodium phosphate, 150 mM sodium chloride, 10% glycerol, 0.5 mg/mL BSA, 0.1% (v/v) DMSO, pH 8 buffer. Ligands were dissolved into DMSO then diluted into buffer immediately prior to the experiment.

**ITC data collection**. All experiments were performed at 30 °C on high-feedback mode with a stirring speed of 806 r.p.m. (Malvern VP-ITC) or 750 r.p.m. (Malvern ITC-200) and a filter time of 1 s. Long-pre-injection delays (~1000 s) were used in order to establish a flat baseline.

Kinetics of inhibition ITC experiments were carried out with POP and TRH in the reaction cell and inhibitor in the syringe. POP was added to the cell solution immediately prior to loading the sample cell and initializing the experiment to avoid depletion of the substrate. Kinetics of initiation ITC experiments were carried out with TRH in the reaction cell and POP/inhibitor in the syringe. The syringe solution was allowed to equilibrate at 30 °C for ~2 h prior to starting the experiment. Ligands were dissolved into DMSO then diluted into buffer immediately prior to the experiment. DMSO was added to the cell solution in order to match the syringe solution to minimize the heats of dilution.

**Baseline correction**. In order to correct for sloped baselines, a common artifact in ITC experiments, a baseline correction procedure was implemented for both kinetics of inhibition and initiation experiments. This procedure involves fitting

lines to the final flat (usually last ~200 s) portion of each injection (Supplementary Fig. 1a). Each of these lines is extrapolated back to the beginning of the respective injection in order to establish a full baseline for each injection. The full baseline is then subtracted from the raw data for each injection to give the baseline-corrected data (Supplementary Fig. 1b).

**Blank subtraction**. ITC injections are accompanied by heat produced due to the dilution of the contents of the syringe into the cell, as well as the mechanical injection process itself. This heat is detected during and immediately following the injection, partially obscuring the desired signal for an amount of time, that depends on the response function of the calorimeter. We have found that the most robust way to circumvent these injection artifacts is to employ a blank experiment performed identically to the actual experiment except without either no enzyme or no substrate. The blank experiment is used to determine when the injection artifact ends ($\tau_{\text{blank}}$) and is used as the starting point for the data analysis of each injection. Note that $\tau_{\text{blank}}$ is approximately given by $\tau_{\text{inj}} + 3 \times \tau_{\text{r}}$, where $\tau_{\text{inj}}$ is the length of the injection and $\tau_{\text{r}}$ is the empirical calorimeter response time ($\approx 10$ s for a Malvern ITC-200 and $\approx 20$ s for a Malvern VP-ITC), although the best estimate is obtained directly from the blank injections (Supplementary Fig. 2).

**ITC data fitting**. POP was assumed to follow Michaelis–Menten kinetics according to:

$$\frac{\mathrm{d}[P]}{\mathrm{d}t} = -\frac{\mathrm{d}[S]}{\mathrm{d}t} = \frac{k_{\text{cat}}[S]([E_0] - [EI])}{[S] + K_{\text{m}}}, \quad (2)$$

where $[P]$, $[S]$, $[E_0]$, and $[EI]$ are the concentrations of product, substrate, total enzyme, and enzyme/inhibitor complex, respectively, $k_{\text{cat}}$ is the catalytic rate constant and $K_{\text{m}}$ is the Michaelis constant. Inhibition was assumed to follow first-order kinetics according to:

$$\frac{\mathrm{d}[EI]}{\mathrm{d}t} = k_{\text{on}}[E][I] - k_{\text{off}}[EI], \quad (3)$$

where $[E]$ is the concentration of free enzyme, and $k_{\text{on}}$ and $k_{\text{off}}$ are the association and dissociation rate constants. The heat flow, $Q(t)$, generated by catalysis was calculated according to:

$$Q(t) = \Delta H_{\text{cat}} V_{\text{cell}} \frac{\mathrm{d}[P]}{\mathrm{d}t}, \quad (4)$$

where $\Delta H_{\text{cat}}$ is the enthalpy of catalysis and $V_{\text{cell}}$ is the volume of the cell, followed by convolution with the empirical instrument response function. The corrected data were fitted using numerical integration of the full set of coupled differential equations, as described in Supplementary Methods. All calculations were performed using MATLAB.

**UV-Vis spectroscopy experiments**. UV-Vis spectroscopy experiments were carried out in triplicate using a BioTek Synergy H4 well plate reader. For the UV-Vis inhibition experiment the solution containing 80 μM carboxybenzyl-Gly-Pro-nitroanilide (ZPG-pN) and 105 nM compound **1** was combined with POP (final concentration 1 nM) ~30 s prior to the first measurement. The absorbance at 405 nm was recorded every 30 s which was converted to a concentration of p-Nitroanaline using the extinction coeffecient at 405 nm (9193 cm$^{-1}$ M$^{-1}$). For UV-Vis initiation experiments POP and compound **1** were preincubated at 277 nM and 1.5 μM respectively for ~2 h hours prior to being diluted to a final concentration of 2.7 and 15 nM into a solution containing 80 μM ZPG-pN ~10 s before the first measurement. The absorbance at 405 nm was recorded every 10 s which was converted to a concentration of p-Nitroanaline using the extinction coeffecient at 405 nm.

**NMR spectroscopy experiments**. NMR experiments were carried out using a Varian 500 MHz spectrometer in a 5 mm probe at 298 K. A DPFGSE sequence was used for water signal suppression. All samples were prepared in 20 mM phosphate buffer (pH 8.0) containing 10% D$_2$O. A 5 mM TRH solution was prepared and POP was added for a final concentration of 50 nM. The first spectrum was acquired 6 min after the addition of compound **5**. Spectra were acquired at each 67 s time interval, 8 transients collected per spectrum. The initial concentrations in the NMR tube were 5 mM TRH, 50 nM POP, and 50 nM ZPP. The peak at ~8 p.p.m. (TRH) was chosen for following the kinetics, as it is was the most resolved.

**Spectroscopy fitting scripts**. All fitting routines were performed using in house MATLAB scripts. Differential equations describing inhibitor association and dissociation and enzyme catalysis were integrated numerically as described in Supplementary Methods.

**Statistical analysis of errors**. Errors were calculated by producing residual sum of squares (RSS) contour plots for each pair of fitted parameters and calculating the confidence level (CL) at each RSS. The errors at the 95% CL are reported (Supplementary Note 1).

**General considerations**. A comprehensive guide to the kinetics of inhibition and initiation techniques is included in Supplementary Note 2.

**Data availability**. The data that support the findings of this study are available from the primary or corresponding author on reasonable request.

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

## Acknowledgements

This research was supported by the Natural Sciences and Engineering Research Council (NSERC, Canada, 327028-09) and the Canadian Institutes of Health Research (CIHR-MOP-136943). J.M.D. was supported by Groupe de Recherche Axé sur la Structure des Protéines and the Drug Development Training Program. C.J.N. was supported by Coordenação de Aperfeiçoamento de Pessoal de Nível Superior (CAPES).

## Author contributions

All authors have given approval to the final version of the manuscript. J.M.D.T., R.O., S.D.C. and C.J.d.N. performed the experiments, J.M.D.T., R.O., S.D.C., C.J.d.N., N.M. and A.K.M. analyzed the data. J.M.D.T., C.J.d.N., N.M. and A.K.M. wrote the manuscript.

## Additional information

**Competing interests:** The authors declare no competing interests.

