## [Peer Review File · Nature Communications]

Reviewers' comments:

Reviewer #1 (Remarks to the Author):

The work by Di Trani and colleagues builds on earlier work in the general field of use of ITC in enzyme kinetics (Todd and Gomez, ref 18, and others) and provides an excellent demonstration of the concept of using ITC for monitoring rate and mechanism of inhibitor binding in the context of enzymatic reactions, a truly exciting new use of this well-established biophysical technique. The study is well-designed and executed, and the manuscript reads well, overall. This demonstration work should be of high degree of interest to many readers of Nature Communications.

I suggest publication after the following issues are dealt with.

- A somewhat major issue. The authors seem to be trying very hard to cast their work as mostly useful in the studies of covalent inhibitors: they keep mentioning this class of inhibitors over and over again, throughout the manuscript, while glossing over the vast majority of inhibitors under consideration for drug development nowadays, namely, the reversible noncovalent kind. The authors need to review and revise their paper accordingly: is the present ITC approach good only for covalent inhibitors or for both covalent and noncovalent types? If the latter, please make this crystal clear, for the sake of the 95%+ of the readers working on noncovalent inhibitors.

- Another somewhat major issue. The paper being about a method of measurement, in general terms, the authors need to provide specifics on the boundaries of usefulness, that is, when/where is the method most applicable, and when/where does it fail (in terms of k_{cat}/K_m for the underlying reaction and k_{on}/k_{off} in terms of inhibitor binding properties). It is understood that the authors are limited in terms of the number and types of enzyme reactions tested for this paper, but at the very least they need to provide some calculations of terminal scenarios of utility and lack thereof. Stated differently, there needs to be some practical advice provided to those who might consider “jumping in” and attempting the technique.

- Related to the comment directly above, this being a paper on a novel methodology, the authors need to provide more details on the practical application of ITC for measuring kinetics of inhibition. For example, the authors need to provide easy-to-follow protocols, along with numerical parameters of ranges of enzyme/substrate/inhibitor concentrations, duration of measurements, etcetera; last but not least, a detailed explanation of how to analyze and interpret the data derived from these protocols is necessary (see relevant comments related to line 83 and

lines 118-120, below). The goal of publication of methodology papers like this is not just to claim that “this can be done” but to also help the community “to do it”.

- Line 12: spell out ITC as it is being used for the first time within the paper (beyond the title itself)

- Lines 16 and 18: spell out POP on line 16 so that it does not come as a surprise on line 18

- Line 58 and elsewhere. The authors refer to K_i as “affinity for enzyme inhibitors”; however, affinity is typically associated with K_d (representing an equilibrium constant of binding/dissociation between two molecules), while K_i is more typically referred to as the inhibitory constant (derived from enzyme kinetics studies, and not representing an equilibrium binding, strictly). Could the authors please clarify the use of terms?

- Line 61: correct “in the PO active,” to “in the POP active site,”??

- Line 73: change “this” to “these”

- Fig 1: define Cbz and Bn

- Line 83: please explain how it was determined that lower values correspond to higher velocities (ie, why is it not the other way around??). This comment relates to the earlier comment demanding a better explanation on the use of the methodology and the associated analyses.

- Line 104, Fig 2a: it is impossible to distinguish “open blue circles” on that plot. Is it even necessary to indicate such given that there is only one dataset plotted in Fig 2a?

- Multiple instances throughout the paper. The compound numbers 1, 2, and so on, are denoted in bold in some places and in regular/plain font elsewhere. The authors need to manually examine the entire manuscript and use bold typeface for compound numbers throughout.

- Lines 118-120. Given the statement that certain data produced from the ITC measurements were “unreliable”, the authors need to provide clear guidelines for when data would be deemed acceptable and when they would not. This comment relates to the earlier comment demanding a better explanation on the use of the methodology and the associated analyses.

- Line 148, fig 3a: please explain ITC-200. Even though it is a well-known instrument by GE Healthcare, it is not acceptable to just drop its name in the caption.

- Line 178, column headers for Table 1: should the k_{on} and k_{off} not contain the “x” in front of $10E5$?

Reviewer #2 (Remarks to the Author):

The manuscript Di Trani and co-workers describes an excellent extension of standard ITC, with potentially very important implication for the study and screening of enzyme inhibitors. The method seems very robust, and the manuscript is very well written. The topic is appropriate for the journal.

I have only a couple of minor concerns:

- 1) The modeling of the observed heat appears to account only for the product formation, but not for the heats of association and dissociation of the inhibitor and substrate to the enzyme. It would be useful to either describe in more detail while the latter are negligible, or, if they are not negligible, to take them into account. Having the model of the species populations as a function of time should easily allow their incorporation, if necessary.
- 2) It would be useful to describe at least some of the details of the modeling (e.g., treatment of baselines and injection artifacts) and the basic equations in the main paper.
- 3) Similarly, in my view seeing only one set of exemplary data is too little to get a good idea of how the method performs. More examples should be included in the main paper.
- 4) It seems the two titration experiments should both report on ΔH_{react} (supp eq. 29), in addition to the kinetic parameters. It would be informative to extend the Table 1 and report the best-fit values, so that the reader can better compare consistency.
- 5) Are there systematic deviations between the inhibition and initiation titrations, for some experimental reason, or could the traces be modeled globally?
- 6) The fits in Figure 3b and d are not at all convincing. 3B looks like a multiphasic decay, whereas the region of curvature in 3D is experimentally not well detected. I am wondering if different concentration conditions could provide clearer confirmation.
- 7) The use of the co-variance matrix for error estimation is outdated and well-known to frequently underestimate the true errors. A more reliable method, such as tracing a contour line in the error surface, would usually give more believable results.

Responses to Reviewers:

Reviewer #1 (Remarks to the Author):

The work by Di Trani and colleagues builds on earlier work in the general field of use of ITC in enzyme kinetics (Todd and Gomez, ref 18, and others) and provides an excellent demonstration of the concept of using ITC for monitoring rate and mechanism of inhibitor binding in the context of enzymatic reactions, a truly exciting new use of this well-established biophysical technique. The study is well-designed and executed, and the manuscript reads well, overall. This demonstration work should be of high degree of interest to many readers of Nature Communications.

I suggest publication after the following issues are dealt with.

- A somewhat major issue. The authors seem to be trying very hard to cast their work as mostly useful in the studies of covalent inhibitors: they keep mentioning this class of inhibitors over and over again, throughout the manuscript, while glossing over the vast majority of inhibitors under consideration for drug development nowadays, namely, the reversible noncovalent kind. The authors need to review and revise their paper accordingly: is the present ITC approach good only for covalent inhibitors or for both covalent and noncovalent types? If the latter, please make this crystal clear, for the sake of the 95%+ of the readers working on noncovalent inhibitors.

This approach is absolutely applicable to all kinds of inhibitors, both covalent and non-covalent. We are grateful to this reviewer for pointing out that the original manuscript gave the misleading impression that the techniques were mainly directed towards covalent inhibitors, as this was certainly not our intention. We have revised the abstract and introduction to ensure that the generality of the approach is not overshadowed by our proof-of-principle application to covalent inhibition. We have also added statements re-emphasizing that the approach applies to both non-covalent and covalent inhibitors and pointed out that two of the molecules successfully studied here bind non-covalently. The abstract now reads:

Although drug development typically focuses on binding thermodynamics, recent studies suggest that kinetic properties can strongly impact a drug candidate's efficacy. Robust techniques for measuring inhibitor association and dissociation rates are therefore essential. To address this need, we have developed a pair of complementary isothermal titration calorimetry (ITC) techniques for measuring the kinetics of enzyme inhibition. The advantages of ITC over standard techniques include speed, generality, and versatility; ITC also measures the rate of catalysis directly, making it ideal for quantifying rapid, inhibitor-dependent changes in enzyme activity. We used our new methods to study reversible covalent and non-covalent inhibitors of prolyl oligopeptidase (POP). We extracted kinetics spanning three orders of magnitude, including those too rapid for standard methods, and measured sub-nM binding affinities, below the typical ITC limit. These results shed light on the inhibition of POP and demonstrate the general utility of ITC-based enzyme inhibition kinetic measurements.

The text on line 194 now reads:

It must be emphasized that, although we have focused on covalent inhibition here, these ITC techniques can be equally well applied to non-covalent inhibitors, for instance compounds **4** and **2**.

- Another somewhat major issue. The paper being about a method of measurement, in general terms, the authors need to provide specifics on the boundaries of usefulness, that is, when/where is the method most applicable, and when/where does it fail (in terms

of k_{cat}/K_m for the underlying reaction and k_{on}/k_{off} in terms of inhibitor binding properties). It is understood that the authors are limited in terms of the number and types of enzyme reactions tested for this paper, but at the very least they need to provide some calculations of terminal scenarios of utility and lack thereof. Stated differently, there needs to be some practical advice provided to those who might consider “jumping in” and attempting the technique.

- Related to the comment directly above, this being a paper on a novel methodology, the authors need to provide more details on the practical application of ITC for measuring kinetics of inhibition. For example, the authors need to provide easy-to-follow protocols, along with numerical parameters of ranges of enzyme/substrate/inhibitor concentrations, duration of measurements, etcetera; last but not least, a detailed explanation of how to analyze and interpret the data derived from these protocols is necessary (see relevant comments related to line 83 and lines 118-120, below). The goal of publication of methodology papers like this is not just to claim that “this can be done” but to also help the community “to do it”.

We feel that this is an excellent suggestion, as there are a large number of factors to consider when determining whether or not a particular system is amenable to this approach and when selecting experimental parameters. Providing additional guidance for non-ITC specialists is essential if the techniques are to be used by a broader community. To this end, we have performed some approximate calculations to delineate the ranges of applicability of these experiments and assembled a guide for researchers wishing to try these experiments in their own laboratories. This information is given in a 14-page document entitled “General Considerations” included in the Supplementary Information. Briefly, this section deals with setting injection volumes and spacing, upper and lower rate limits and their dependence on K_m , k_{cat} and ΔH_{cat} , maximum and minimum K_i values, optimizing signal, choosing enzyme, substrate, and inhibitor concentrations, and data collection and analysis workflows. We also report the approximate limiting values of k_{on} , k_{off} , and K_i in the manuscript. The text on line 196 now reads:

We estimate that inhibitor residence times from roughly 10 seconds to 30 minutes, association rates from 10^3 to 10^7 $M^{-1} s^{-1}$, and K_i values down to about 1 pM are accessible using this approach under typical conditions (see Supplementary Information: General considerations; for details and a practical guide to use of these techniques).

- Line 58 and elsewhere. The authors refer to K_i as “affinity for enzyme inhibitors”; however, affinity is typically associated with K_d (representing an equilibrium constant of binding/dissociation between two molecules), while K_i is more typically referred to as the inhibitory constant (derived from enzyme kinetics studies, and not representing an equilibrium binding, strictly). Could the authors please clarify the use of terms?

In order to avoid confusion, we have changed the terminology throughout such that K_i is referred to as the inhibition constant rather than the affinity constant. We do point out, however, that K_i is usually taken to be equal to the equilibrium binding constant of the inhibitor. The text at line 51 now reads:

Here we present a pair of rapid, complementary ITC methods that simultaneously measure inhibitor association and dissociation rates and the inhibitory constant K_i , for enzyme inhibitors in an hour or less.

The text at line 213 now reads:

In order to obtain reliable information, the concentration of enzyme in the ITC cell is ideally no more than 500-fold the equilibrium dissociation constant, which is generally taken as equal to the K_i .

- **Line 12: spell out ITC as it is being used for the first time within the paper (beyond the title itself)**
- **Lines 16 and 18: spell out POP on line 16 so that it does not come as a surprise on line 18**
- **Line 61: correct "in the PO active," to "in the POP active site,"??**
- **Line 73: change "this" to "these"**
- **Fig 1: define Cbz and Bn**
- **Multiple instances throughout the paper. The compound numbers 1, 2, and so on, are denoted in bold in some places and in regular/plain font elsewhere. The authors need to manually examine the entire manuscript and use bold typeface for compound numbers throughout.**
- **Line 178, column headers for Table 1: should the k_{on} and k_{off} not contain the "x" in front of $10E5$?**

These corrections have been made.

- **Line 83: please explain how it was determined that lower values correspond to higher velocities (ie, why is it not the other way around??). This comment relates to the earlier comment demanding a better explanation on the use of the methodology and the associated analyses.**

We agree with the reviewer that this important point would likely not be clear to a non-ITC specialist. We have added a brief explanation in the manuscript and a more detailed discussion of determining enzyme catalytic parameters in the General Considerations section. The text at line 81 now reads:

In both experiments, the heat flow or power (y-axis) is plotted as a function of time (x-axis). Exothermic and endothermic reactions deflect the ITC signal downwards and upwards, respectively. The power is linearly related to the enzyme velocity, with larger deflections corresponding to higher velocities. In the kinetics of inhibition experiments, the ITC cell contained POP and the substrate thyrotropin releasing hormone (TRH). POP cleaves TRH producing the free acid form of TRH and ammonia as well as heat, which is detected by the ITC instrument (See Supplementary Information: General Considerations). The rate of catalysis was initially constant, giving a horizontal line. Compounds **1** and **4** were added to the cell in a series of three (Fig. 2a) and six (Fig. 2c) injections. In each case, the enzyme was increasingly inhibited and the power values shifted upward, since the rate of (exothermic) catalysis and downward deflection was reduced after each injection.

- **Line 104, Fig 2a: it is impossible to distinguish "open blue circles" on that plot. Is it even necessary to indicate such given that there is only one dataset plotted in Fig 2a?**

The reference to "open blue circles" has been removed.

- Lines 118-120. Given the statement that certain data produced from the ITC measurements were “unreliable”, the authors need to provide clear guidelines for when data would be deemed acceptable and when they would not. This comment relates to the earlier comment demanding a better explanation on the use of the methodology and the associated analyses.

This is an excellent example of where the General Considerations discussion is useful. Rather than simply referring to the analysis as “unreliable”, we can now reference the section in the Supplementary Information where the limits of the technique are explained. The text at line 126 now reads:

For compounds **3** and **5**, dissociation occurred very slowly compared to the rate of ITC baseline drift and k_{off} could not be accurately measured using the initiation experiment (See Supplementary Information: General considerations).

- Line 148, fig 3a: please explain ITC-200. Even though it is a well-known instrument by GE Healthcare, it is not acceptable to just drop its name in the caption.

We have modified all references in the figure legends to “Malvern ITC-200 calorimeter” and “Malvern VP-ITC calorimeter”

Reviewer #2 (Remarks to the Author):

The manuscript Di Trani and co-workers describes an excellent extension of standard ITC, with potentially very important implication for the study and screening of enzyme inhibitors. The method seems very robust, and the manuscript is very well written. The topic is appropriate for the journal.

I have only a couple of minor concerns:

1) The modeling of the observed heat appears to account only for the product formation, but not for the heats of association and dissociation of the inhibitor and substrate to the enzyme. It would be useful to either describe in more detail while the latter are negligible, or, if they are not negligible, to take them into account. Having the model of the species populations as a function of time should easily allow their incorporation, if necessary.

This is an important detail that was not given enough attention in the original manuscript. We had assumed that the low concentrations of enzyme ensured that the heat released by binding would be negligible, but we had not tested the assumption. We therefore measured the heat of binding for a representative inhibitor under high enzyme concentration conditions and extrapolated the results to the conditions of our kinetic assays. Indeed, the heats of binding are much less (200-fold) than the baseline noise and therefore can be safely ignored. These calculations are included in the Supplementary Information as follows:

Enthalpy of binding in kinetics of inhibition and initiation experiments. In the kinetics of inhibition and initiation experiments performed here, the instrument measures both the heat generated by enzymatic catalysis and the heat produced due direct interaction of the inhibitor with the enzyme. In order to estimate the relative magnitudes of these two effects, the enthalpy of binding was directly measured using an ordinary ITC binding experiment at a

concentration of POP significantly higher (approx. 100 to 1000-fold greater, 5 μM) than those used in kinetics of inhibition and initiation experiments (Supplementary Fig. 5). Binding data were then modelled at the concentrations of enzyme and inhibitor and injection size and spacing used for the kinetic experiment for compound **5**, with representative instrumental noise estimated from the baselines, ΔH_{bind} taken from Supplementary Table 2 and K_i taken as a previously reported value (0.5 nM)¹³ (Supplementary Fig. 6).

Supplementary Figure 5. Compound **5** prolyl oligopeptidase binding experiments. Compound **5** (90 μM) was titrated into 5 μM POP in cell using a Malvern ITC-200 calorimeter. First injection is 0.1 μL (not shown) and subsequent 8 injections are 4.3 μL . Molar ratio of compound **5**/POP vs Total Heat for each injection peak (open black circles) with fit (red line).

Supplementary Table 2. Summary of compound **5** prolyl oligopeptidase binding parameters.

ΔH_{bind}	$-22.2 \pm 0.1 \text{ Kcal mol}^{-1}$
N	0.93 ± 0.03
K_i	--

Supplementary Figure 6. Modelling heat of binding in kinetics of inhibition experiments. Modelled kinetics of inhibition experiment with compound **5** using only contributions from the heats of binding.

It is apparent that at the concentrations used in this study the heats of binding are negligible and can be omitted from the analysis. We estimate that heats of binding only become significant ($>2\times$ the noise) if either the binding enthalpy or POP concentration are over 200-fold greater than what was measured/used in this study.

They are referred to in the manuscript on line 102:

Note that concentrations of enzyme are so low in these experiments ($\approx 10\text{nM}$) that ITC detects only heats of catalysis, while heats of inhibitor/enzyme binding can be safely ignored (See Supplementary methods).

2) It would be useful to describe at least some of the details of the modeling (e.g., treatment of baselines and injection artifacts) and the basic equations in the main paper.

We agree that this is important that readers are able to access this information easily. We have modified the Methods section in the manuscript which now reads (starting on line 244):

Baseline correction. In order to correct for sloped baselines, a common artifact in ITC experiments, a baseline correction procedure was implemented for both kinetics of inhibition and initiation experiments. This procedure involves fitting lines to the final flat (usually last ~200 s) portion of each injection (Supplementary Fig. 1a). Each of these lines is extrapolated back to the beginning of the respective injection in order to establish a full baseline for each injection. The full baseline is then subtracted from the raw data for each injection to give the baseline-corrected data (Supplementary Fig. 1b).

Blank subtraction. ITC injections are accompanied by heat produced due to the dilution of the contents of the syringe into the cell as well as the mechanical injection process itself. This heat is detected during and immediately following the injection, partially obscuring the desired signal for an amount of time that depends on the response function of the calorimeter. We have found that the most robust way to circumvent these injection artifacts is to employ a blank experiment performed identically to the actual experiment except without either no enzyme or no substrate. The blank experiment is used to determine when the injection artifact ends (τ_{blank}) and is used as the starting point for the data analysis of each injection. Note that τ_{blank} is approximately given by $\tau_{\text{inj}} + 3 \times \tau_r$, where τ_{inj} is the length of the injection and τ_r is the empirical calorimeter response time (≈ 10 seconds for a Malvern ITC-200 and ≈ 20 seconds for a Malvern VP-ITC), although the best estimate is obtained directly from the blank injections (Supplementary Fig. 2).

ITC data fitting. POP was assumed to follow Michaelis Menten kinetics according to:

$$\frac{d[P]}{dt} = -\frac{d[S]}{dt} = \frac{k_{\text{cat}}[S]([E_0] - [EI])}{[S] + K_m}, \quad (2)$$

where [P], [S], [E₀], and [EI] are the concentrations of product, substrate, total enzyme, and enzyme/inhibitor complex, respectively, k_{cat} is the catalytic rate constant and K_m is the Michaelis constant. Inhibition was assumed to follow first-order kinetics according to:

$$\frac{d[EI]}{dt} = k_{\text{on}}[E][I] - k_{\text{off}}[EI] \quad (3)$$

where [E] is the concentration of free enzyme, and k_{on} and k_{off} are the association and dissociation rate constants. The heat flow, Q(t), generated by catalysis was calculated according to:

$$Q(t) = \Delta H_{\text{cat}} V_{\text{cell}} \frac{d[P]}{dt} \quad (4)$$

where ΔH_{cat} is the enthalpy of catalysis and V_{cell} is the volume of the cell, followed by convolution with the empirical instrument response function. The corrected data were fitted using numerical integration of the full set of coupled differential equations, as described in the Supplementary methods. All calculations were performed using MATLAB.

3) Similarly, in my view seeing only one set of exemplary data is too little to get a good idea of how the method performs. More examples should be included in the main paper.

We have added 4 additional panels to Figure 2 in order to give a better sense of the performance of the method. The new figure is shown below:

Fig. 2. Kinetics of inhibition and initiation experiments. a) Kinetics of inhibition experiment with compound 1 using a Malvern ITC-200 calorimeter. Compound 1 is titrated into cell containing POP and TRH. b) Overlay of injections 2, 3 and 4 (orange, yellow and purple circles) with fit (black line). c) Kinetics of inhibition experiment with compound 4 using a Malvern ITC-200 calorimeter. Compound 4 is titrated into cell containing POP and TRH. d) Overlay of injections 2, 4, 5, 7 (orange, yellow, purple and green circles) with fit (black line). e) Kinetics of initiation experiment of compound 1 with a Malvern VP-ITC calorimeter. Compound 1 and POP are titrated in the cell containing TRH. f) Overlay of injections 2, 3, 4 and 5 (orange, yellow, purple and green circles) with fit (black lines). g) Kinetics of initiation experiment with compound 4 using a Malvern ITC-200 calorimeter. Compound 4 and POP are titrated into cell containing TRH. h) Overlay of injections 2, 4, 5, 7 (orange, yellow, purple and green circles) from C with fit (black lines).

4) It seems the two titration experiments should both report on ΔH_{react} (supp eq. 29), in addition to the kinetic parameters. It would be informative to extend the Table 1 and report the best-fit values, so that the reader can better compare consistency.

We agree that this is a good way to judge the consistency of the analyses. We have added fitted ΔH_{cat} parameters to Table 1 as follows:

Table 1. Kinetic and thermodynamic parameters.

Cpd	Experiment	$k_{\text{on}} \times 10^5 \text{ M}^{-1} \text{ s}^{-1}$	$k_{\text{off}} \times 10^{-4} \text{ s}^{-1}$	K_i nM	${}^b \Delta H_{\text{cat}} \text{ kcal mol}^{-1}$
1	ITC Inhib.	1.12 ± 0.02	$43.1^a \pm 0.9$	38.4 ± 0.4	-6.69 ± 0.02
	ITC Init.	$1.21^a \pm 0.01$	82.1 ± 0.4	58.7 ± 0.5	-5.86 ± 0.01
	UV-VIS	1.43 ± 0.02	31.4 ± 0.3	$947^a \pm 2$	
2	ITC Inhib.	--	--	597 ± 28	-6.79 ± 0.03
	ITC Init.	--	--	744 ± 16	-10.88 ± 0.06
3	ITC Inhib.	155.5 ± 0.4	$9^a \pm 1.5$	0.063 ± 0.01	-9.91 ± 0.03
4	ITC Inhib.	44 ± 1	$249^a \pm 7$	5.6 ± 0.1	-5.33 ± 0.03
	ITC Init.	$20^a \pm 1$	261 ± 9	13.1 ± 0.4	-7.19 ± 0.04
5	ITC Inhib.	4.18 ± 0.04	< 1.05	< 2.5	-7.97 ± 0.01
	NMR	5.4 ± 0.7	--	--	
	Lit ²⁸	0.7	3	0.5	

^acalculated using $K_i = k_{\text{off}}/k_{\text{on}}$

^b ΔH_{cat} are floated in order account for uncertainties in enzyme concentration as the magnitude of the heat flow is directly proportional to $[\text{Eo}]\Delta H_{\text{cat}}$.

5) Are there systematic deviations between the inhibition and initiation titrations, for some experimental reason, or could the traces be modeled globally?

This is an excellent point. Indeed global fitting of inhibition and initiation experiments should be possible. We have performed such analysis for compounds **1**, **2** and **4** and included the results in the Supplementary Information. The global fits give reasonably good agreement, although not as close as the individual fits, and the extracted parameters are close to those of the individual fits. We have modified the manuscript to refer to these results as follows:

Global fits of combined initiation and inhibition datasets are in good agreement with experiment signals and yield similar kinetic and thermodynamic parameters to the individual fits (See Supplementary results).

The global fits are as follows:

6) The fits in Figure 3b and d are not at all convincing. 3B looks like a multiphasic decay, whereas the region of curvature in 3D is experimentally not well detected. I am wondering if different concentration conditions could provide clearer confirmation.

We agree with the reviewer that the spectroscopic data in Figure 3d could be substantially improved. We repeated the experiments and the early curvature in the data is far better defined, showing improved agreement with the ITC-derived kinetics, as follows:

In the case of Figure 3b, the appearance of a biphasic decay results from the non-zero slope for the latter half of the data points which is due to residual uninhibited enzyme. If we had added more inhibitor, the data would have more clearly resembled a single exponential decay with a horizontal asymptote. This was not possible as increasing the concentration of inhibitor to such an extent would have made the initial decay too rapid to measure. This nicely illustrates the challenges of measuring inhibition kinetics using conventional methods. We have added a brief discussion of this point at line 145:

Note that the final non-zero slope of the curve in Fig. 3b is due to remaining uninhibited POP. Increasing the amount of **5** added would make this portion of the curve more horizontal, but would also increase the rate of inhibition (early curvature) beyond values accessible by this method, highlighting some of the challenges of measuring inhibition kinetics by non-ITC methods.

7) The use of the co-variance matrix for error estimation is outdated and well-known to frequently underestimate the true errors. A more reliable method, such as tracing a contour line in the error surface, would usually give more believable results.

We agree that a detailed analysis of the error surface provides a robust analysis of uncertainties in extracted parameters. We have followed the reviewer's excellent suggestion to define confidence bounds in terms of the error surface and updated Table 1 as follows:

Table 1. Kinetic and thermodynamic parameters.

Cpd	Experiment	$k_{on} \times 10^5 \text{ M}^{-1} \text{ s}^{-1}$	$k_{off} \times 10^{-4} \text{ s}^{-1}$	$K_i \text{ nM}$	${}^b \Delta H_{cat} \text{ kcal mol}^{-1}$
1	ITC Inhib.	1.12 ± 0.02	$43.1^a \pm 0.9$	38.4 ± 0.4	-6.69 ± 0.02
	ITC Init.	$1.21^a \pm 0.01$	82.1 ± 0.4	58.7 ± 0.5	-5.86 ± 0.01
	UV-VIS	1.43 ± 0.02	31.4 ± 0.3	$947^a \pm 2$	
2	ITC Inhib.	--	--	597 ± 28	-6.79 ± 0.03
	ITC Init.	--	--	744 ± 16	-10.88 ± 0.06
3	ITC Inhib.	155.5 ± 0.4	$9^a \pm 1.5$	0.063 ± 0.01	-9.91 ± 0.03
4	ITC Inhib.	44 ± 1	$249^a \pm 7$	5.6 ± 0.1	-5.33 ± 0.03
	ITC Init.	$20^a \pm 1$	261 ± 9	13.1 ± 0.4	-7.19 ± 0.04
5	ITC Inhib.	4.18 ± 0.04	< 1.05	< 2.5	-7.97 ± 0.01
	NMR	5.4 ± 0.7	--	--	
	Lit ²⁸	0.7	3	0.5	

^acalculated using $K_i = k_{off}/k_{on}$

^b ΔH_{cat} are floated in order account for uncertainties in enzyme concentration as the magnitude of the heat flow is directly proportional to $[Eo]\Delta H_{cat}$.

We note that we have used 95% confidence bounds in the new analysis which are larger than the standard deviations reported in the earlier version of the manuscript. The details of the procedure and error surface plots are given in the Supplementary Information.

Reviewers' comments:

Reviewer #1 (Remarks to the Author):

The authors have done a very good job at responding to all reviewers' critiques. I recommend publication of the so-revised manuscript.

Reviewer #2 (Remarks to the Author):

In my view the authors have addressed thoroughly all concerns raised.